# State Financial Losses in Public Procurement Construction Projects in Indonesia

**Herry Ludiro Wahyono [1,*], Jati Utomo Dwi Hatmoko [2] and Rizal Z. Tamin [3]**

[1]   Department of Civil Engineering, Faculty of Engineering, State Polytechnic of Semarang 50275, Indonesia
[2]   Department of Civil Engineering, Faculty of Engineering, Diponegoro University, Semarang 50275, Indonesia; jati.hatmoko@gmail.com
[3]   Institute Technology of Bandung, Bandung 40132, Indonesia; rztamin@gmail.com
[*]   Correspondence: l.herry@yahoo.com or herry.ludiro.wahyono@polines.ac.id; Tel.: +62-8122-802-500

**Abstract:** To sustainably eradicate corruption, extraordinary efforts are needed particularly in the context of construction projects. In this study, the state losses in government-funded construction projects in Indonesia is analyzed. Research data were collected through field observations for types of work that were not completed per technical specifications. Data analysis was performed using descriptive statistics. The results indicate that structural work was responsible for 98.70% of total state losses. The top three state losses were found to be associated with concrete, roads, and front stonework, and the values are 40.40%, 28.43%, and 18.23% respectively. Loss in architectural work was the smallest (1.30%). The outcomes of this survey offer a knowledge basis for law enforcement authorities and construction project supervisors to pay more attention to projects involving concrete structures, rigid pavement structures during road construction, and stonework.

**Keywords:** state financial losses; infrastructure project; corruption

---

## 1. Introduction

Corruption in construction projects is widespread. It occurs in a number of developing countries in Asia, such as China, India, and Indonesia. Additionally, corruption also occurs in South Africa and Ghana on the African continent. Since 2002, the government of the Republic of Indonesia (RI) has been more seriously committed to eradicating corruption. However, since then, corruption has not been eradicated, and the plan has not been optimally implemented. There is an urgent need for more effectual measures to intensively and sustainably halt professional criminal acts of corruption, because corruption has harmed the country's finances and economy, as well as negatively impacted its national development. Moreover, most government institutions that handle corruption cases are not operating effectively or efficiently in combating corruption. On this basis, and within the context of the eradication of corruption crimes stipulated in the 1999 corruption eradication law [1] and its refinement [2], a commission for eradicating the criminal act of corruption was formed. The Corruption Eradication Commission (KPK) is an independent body with its main aim being to combat corruption throughout Indonesia [3].

The seriousness of the Indonesian government in eradicating corruption is evidenced by the establishment of the Corruption Court in the Central Jakarta District Court [4,5] and in 33 district courts in the provincial capital throughout the Republic of Indonesia [6–8]. Anti-corruption institutions in Indonesia, particularly KPK, are prioritizing efforts to prosecute corruption rather than focusing on prevention [9]. To sustainably eradicate corruption, efforts to repress corruption and to prevent it must be balanced [10]. According to the KPK Annual Report from 2004 to 2017, there was a significant increase in cases of corruption related to the low quality of construction work in government projects. In the aforementioned 14-year period, 202 cases of corruption were claimed to have been submitted,

with a 184% average annual increase [11]. Similar to the annual report of the Supreme Audit Agency (BPK), there were a total of 232 cases of crime, representing an average annual increase of 104% [12]. This indicates that construction project corruption cases in Indonesia were increasing, evidencing the need for a balance between prosecuting and averting corruption.

Criminal acts of corruption may occur in any stage of the implementation phase of a construction project [13]. The following elements constitute corruption in Indonesia: (a) the act is against the law and (b) it enriches oneself or others or corporations or (c) it is detrimental to the country's finances or economy [1,2]. Corruption can be in the form of bribes, fraudulent acts, illegal gratuities, extortion, or blackmail [11]. Corruption in public construction projects in Indonesia can be traced back to the implementation of the budgeting process at the parliament [13].

In China, corruption is caused by a flawed regulatory system, which results in leaders being able to cheat, ineffective minor sanctions, weak supervision, extra-procedural permission processes, nepotism, collusion, and professional misconduct [14]. In South Africa, corruption is related to the following: the involvement of government officials through bribery and tender manipulation, moral factors (minor sanctions, bad ethics, no fear of violation), and methods to eradicate corruption (strict rules and sanctions) [15]. In southwest Nigeria, corruption is more related to poverty, greed, political involvement, incompetence, poor quality, excessive profits, and shamanic practices [16]. Corruption can occur via abuse of power, bribery, extra-procedural permission processes, and contract value mark-up [17,18]. Political connections can exacerbate asymmetric information and anti-corruption measures [19].

Eradicating corruption in Indonesia does not merely involve law enforcement or debate among legal experts, but rather political-business oligarchy fights [20]. Corruption can be minimized by effective top-down supervision [21]. In developing countries, corrupt officials learn from previous corruption cases to create and implement new corruption strategies [18], resulting in non-conformance to the costs and time originally targeted by the projects. [22].

Efforts to overcome corruption in infrastructure procurement in Ghana, which takes the form of conflicts of interest, bribery, embezzlement, tender manipulation, and fraud, have included introducing regulations through the application of the 2003 Public Procurement Act. This has been done in an effort to control procurement, improve social equality, develop trust, guarantee a selection of competent and transparent contractors, and display accountability [23]. The level of corruption in Indonesian local government remains high and has not been affected by the results of audits regarding the accountability of local government financial reports. Although the audit reports have been based on audit opinions, there are weaknesses in the internal control system and with non-compliance with statutory provisions [24–26]. Based on audit results, the number of corruption cases regarding the financial statements of regional governments has continued to increase, despite the opinion of BPK and public opinion [27,28].

Project failure can occur in the form of construction failure or building failure [3,29]. Construction failure occurs when the results of construction work are not of the quality stipulated in the contract, whereas the results of non-functional construction work are classified as building failure. Failures caused by fraudulent acts by one or several parties [30] result in losses to state finances, which include money or goods [31]. Financial losses include shortages of money, securities, and goods that are measured and definite. These occur as a result of illegal actions whether intentional or not, and the real loss of state finances must be determined with legal certainty [32].

China has attempted to prevent the occurrence of criminal acts of corruption in the implementation of construction projects through managing projects, implementing regulations, training construction actors, and imposing strict sanctions [33]. Likewise, to prevent corruption in India, dissemination of information about how to manage projects, implementation of regulations, training of workers, and imposing punitive sanctions on policymakers and the construction industry have contributed to achieving economic goals with efficiency, equitability, and transparency [34]. To date, no technical audit has been completed, so there is a need to identify work items and types of construction that often cause state financial losses. By recognizing such work items, the construction service industry players

can pay more attention to them during the implementation process, thus minimizing the chances of state financial losses.

Therefore, the anti-corruption institutions in Indonesia need to identify areas in which construction projects often cause considerable state financial losses, so they can focus their attention on assisting in these areas. Thus, the construction service industry players in these areas can jointly minimize the opportunity for errors in the implementation of construction work that can lead to state financial losses. Anti-corruption institutions and stakeholders of the construction service industry also need to identify the most frequent and/or the greatest number of cases in the fiscal year that cause losses to the state. With this knowledge, further research can be conducted to identify and analyze in depth why the number of cases and the amount of state financial losses have increased in the relative fiscal years. The aim of the study was to analyze the financial losses in government-funded construction projects in Indonesia. The identification process was completed by analyzing events based on a time series of public building projects according to work items, types of work, and percentage losses of state finances from the audit results of buildings in several regions in Indonesia.

## 2. Research Method

As the research sample, data were collected using field observations of construction projects from the results of research requests made by the auditor institution and investigation institutions in Indonesia. Determination of the research sample was not random but chosen: construction projects that were indicated to cause state financial losses in their implementation were selected. Corruption in the context of construction project is considered to occur when the works are not finished according to the contract in terms of quality and/or volume of the works resulting in state financial losses.

Research data were collected over 11 years from 2003 to 2014 at the request of the auditor or investigator as outlined in the technical test research report. The results of the data collection included 64 projects, 12 cities/districts, 9 budget years (2003, 2004, 2007, 2008, 2009, 2011, 2012, 2013, and 2014), 15 work items, and 2 types of work (architecture and structure) (Table 1).

**Table 1.** Profile of research data in the study area.

| Criteria | Total | Central Java | Jakarta | East Kalimantan |
|---|---|---|---|---|
| Project (amount of case) | 64 | 56 | 4 | 4 |
| City/regency (location) | 12 | 10 | 1 | 1 |
| Fiscal year | 9 | 9 | 2 | 3 |
| Work item | 15 | 15 | 15 | 15 |
| Type of work | 2 | 2 | 2 | 2 |

Data from Tabel 1 shown that 64 projects were obtained via the following seven stages: (1) collecting contract documents; (2) gathering field implementation documents obtained from the auditor/investigator research requests; (3) engaging in initial discussions with the auditor/investigator; (4) conducting field surveys; (5) discussing the data collected by the investigators to determine the field inspection strategy and technical testing; (6) implementing field inspection and technical testing at the project site and in the laboratory and having data on examination and technical testing, which were outlined in the minutes of the examination, signed by the auditor/investigator, researcher, owner, supervisory consultant, and contractor; and (7) analyzing secondary data and primary data for information regarding volume, quantity or quality, and costs to create research reports.

The data from the 64 projects were analyzed using descriptive statistical methods. Regarding the respondents' confidentiality, the name of the project and the city/district where the project was located were disguised. The project name was replaced with a "Case-n" identity and project location with "Location-n".

## 3. Results

The data in this study were analyzed using descriptive statistical methods. The descriptive statistics included the following: frequency, average value, smallest value, and greatest value. Table 2 lists the areas that had the most cases and the regions that had the greatest financial losses. Table 2 provides the location of each project/case, the number of cases in the location and the corresponding rank, and the amount of state financial losses and the corresponding rank.

**Table 2.** Number of cases and state financial losses by location.

| Location of Cases | Cases | | State Financial Losses | |
| --- | --- | --- | --- | --- |
| | Number of Projects | Rank | IDR (Rupiah) | Rank |
| Location-1 | 10 | 3 | 1,012,447,206 | 5 |
| Location-2 | 2 | 10 | 514,000,355 | 7 |
| Location-3 | 5 | 5 | 95,006,055 | 8 |
| Location-4 | 1 | 11 | 340,000 | 11 |
| Location-5 | 6 | 4 | 63,562,917 | 9 |
| Location-6 | 13 | 1 | 1,843,462,159 | 1 |
| Location-7 | 4 | 6 | 1,333,095,016 | 3 |
| Location-8 | 3 | 8 | 764,800,764 | 6 |
| Location-9 | 4 | 7 | 1,739,597,846 | 2 |
| Location-10 | 1 | 12 | 269,000 | 12 |
| Location-11 | 3 | 9 | 59,788,939 | 10 |
| Location-12 | 12 | 2 | 1,130,877,372 | 4 |
| Total | 64 | | 8,557,247,629 | |
| Average value | 5 | | 713,103,969 | |
| Smallest value | 1 | | 269,000 | |
| Greatest value | 13 | | 1,843,462,159 | |

As can be seen in Table 2, the total number of projects/cases was 64, with an average of five cases for each location. The locations that had the fewest cases (1) were Location-4 and Location-10, whereas Location-6 had the most cases (13). The total amount of state financial losses was IDR 8,557,247,629 with an average state financial loss of IDR 713,103,969 for each location. The location that had the least financial loss (IDR 269,000) was Location-10, whereas Location-6 had the largest financial loss (IDR 1,843,462,159).

Table 3 identifies the fiscal year (TA) that occurred most frequently in the cases and the fiscal year in which the largest state financial loss occurred. It provides the number of cases/projects in the fiscal year and the corresponding ranking, and the amount of state financial losses and the corresponding ranking. As can be seen, there was an average of 7.1 cases for each fiscal year. TA 2004 was the fiscal year with the fewest cases (2), whereas TA 2012 had the most (13).

**Table 3.** Number of cases and state financial losses according to the project budget year.

| Budget Year | Cases | | State Financial Losses | |
| --- | --- | --- | --- | --- |
| | No. of Projects | Rank | IDR (Rupiah) | Rank |
| 2003 | 4 | 7 | 60,057,939 | 9 |
| 2004 | 2 | 9 | 109,069,220 | 7 |
| 2007 | 10 | 3 | 1,339,979,899 | 2 |
| 2008 | 9 | 4 | 978,053,134 | 4 |
| 2009 | 5 | 6 | 813,751,514 | 6 |
| 2011 | 6 | 5 | 63,562,917 | 8 |
| 2012 | 13 | 1 | 3,226,738,516 | 1 |
| 2013 | 3 | 8 | 835,157,118 | 5 |
| 2014 | 12 | 2 | 1,130,877,372 | 3 |
| Total | 64 | | 8,557,247,629 | |
| Average value | 7.1 | | 950,805,292 | |
| Smallest value | 2 | | 60,057,939 | |
| Greatest value | 13 | | 3,226,738,516 | |

The amount of state financial losses incurred in each fiscal year (TA) is displayed in Table 3 and Figure 1, the average of which is IDR 950,805,292 for each budget year. The fiscal year with the least financial loss (IDR 60,057,939) was TA 2003, whereas the largest state financial losses (IDR 3,226,738,516) occurred in TA 2012.

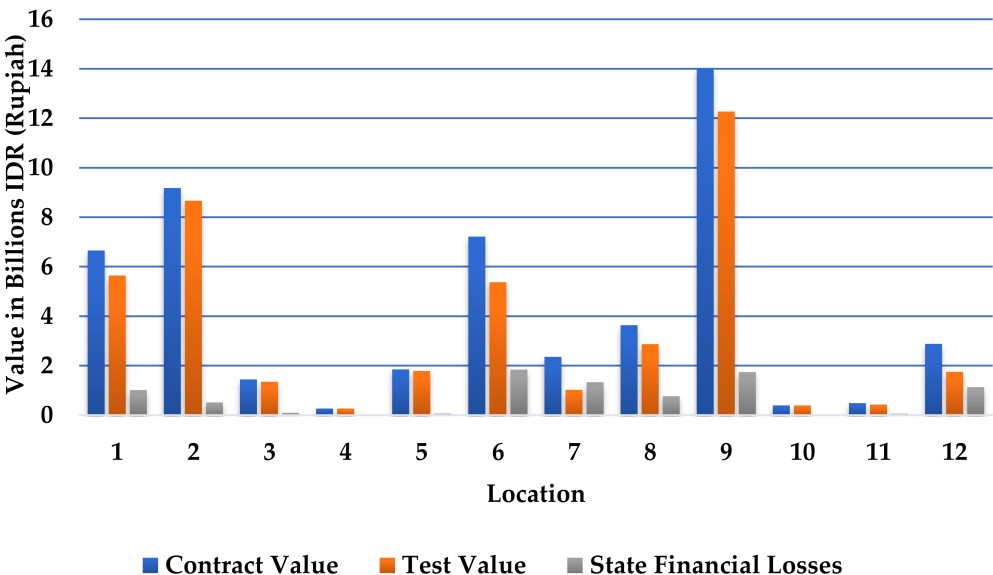

**Figure 1.** Comparison of contract value, test value, and loss of state finance (real value based on site examination).

To identify work items and types of work that caused the largest state financial losses, we analyzed the percentage of state financial losses incurred according to the type of work, as compiled in Table 4. Table 4 lists work items, types of work, and the amount of state financial losses incurred by each item of work. Table 4 is arranged by sorting the amount of state financial losses incurred by each work item from large to small. Table 4 outlines the five items of work that incurred the greatest financial losses: concrete works (IDR 3,457,181,412 or 40.401%), road work (IDR 2,432,428,310 or 28.425%), pair of stone splits (IDR 1,559,838,741 or 18,228%), land works (IDR 720,355,120 or 8.418%), and gabion (IDR 148,574,660 or 1.736%). Combined, these five work items were responsible for 97.21% of the total state financial losses incurred.

**Table 4.** Total state financial losses according to job items.

| No. | Work item | Deviation from contract (IDR) | | Value (IDR) | Percentage |
|---|---|---|---|---|---|
| | | Architecture | Structural | | |
| 1 | Concrete work | - | 3,457,181,412 | 3,457,181,412 | 40.401% |
| 2 | Street work | - | 2,432,428,310 | 2,432,428,310 | 28.425% |
| 3 | Stone work | - | 1,559,838,741 | 1,559,838,741 | 18.228% |
| 4 | Soil work | - | 720,355,120 | 720,355,120 | 8.418% |
| 5 | Gabion | - | 148,574,660 | 148,574,660 | 1.736% |
| 6 | Galvanized pipe and threaded iron | 66,656,207 | - | 66,656,207 | 0.779% |
| 7 | Woodwork | 10,511,300 | 52,984,004 | 63,495,304 | 0.742% |
| 8 | Boulders | - | 43,488,568 | 43,488,568 | 0.508% |
| 9 | Doors and windows work | 28,053,474 | - | 28,053,474 | 0.328% |
| 10 | Steelwork | - | 21,862,376 | 21,862,376 | 0.255% |
| 11 | Tile work | 7,635,557 | - | 7,635,557 | 0.089% |
| 12 | Paving block | - | 5,828,700 | 5,828,700 | 0.068% |
| 13 | Curb (the edging of pavement) | - | 1,240,200 | 1,240,200 | 0.014% |
| 14 | Iron door leaf | 340,000 | - | 340,000 | 0.004% |
| 15 | Fabric chair | 269,000 | - | 269,000 | 0.003% |
| | Total | 113,465,537 | 8,443,782,092 | 8,557,247,629 | 100.000% |
| | Percentage | 1.30% | 98.70% | - | - |

## 4. Discussion

As shown in Table 2, Location-6 had the most cases and incurred the largest state financial losses. Location-10 had the fewest cases and incurred the smallest financial losses in the case study location. The frequent cases indicated the highest state financial losses rather than the rare cases. The symptom of corruption in the public building projects can be captured by these cases. However, in terms of efforts to reduce the number of cases and simultaneously reduce the amount of state losses, the construction projects at Location-6 require tighter supervision.

Table 3 shows that there was a pattern of corruption: the fiscal year of 2012 has the most cases (i.e., 13 cases) also experienced the biggest loss about IDR 3,226,738,516. This indicates that there is a significant relationship between the number of cases that occur and the amount of state losses. This is similar to findings reported from China, stating that corruption can occur due to mega projects, number of projects, complexity of work, and the competence of construction actors [14].

As can be seen in Table 4, structural work incurred more losses than architectural work. The overall ratio was 98.70: 1.30 for structural work to architectural work. The top five work items that were responsible for the state financial losses were in the category of structural work, responsible for 97.21% of the overall state financial loss. Thus, it can be concluded that structural work needs more serious attention and tighter supervision than architectural work.

## 5. Conclusions

This study provided a financial loss analysis of the building projects in different provinces in Indonesia, especially those in Central Java, Jakarta and East Kalimantan. Of the 64 projects examined during the nine fiscal years, the largest financial loss occurred in 2012, with a loss of IDR 3,226,738,516, and the area with the most cases was Location-6, in the area of Central Java Province.

The construction types that incurred the largest losses were structural work which caused 98.70% of the losses to state finances, comprising 40.40% concrete work items, 28.43% road works, 18.23% split stonework jobs, and 8.42% earthworks. The rest (1.30%) of the losses were caused by architectural work. Consequently, the structural work projects must receive more attention and be more strictly supervised than other jobs, especially with respect to concrete work, road pavement, split stone pairs, and earthworks. To complement and to further comprehend the results of this study, future research can be carried out on internal and external stakeholder mapping to assist with projects' quality control processes.

**Author Contributions:** Conceptualization, H.L.W. and R.Z.T.; methodology and investigation, H.L.W. and J.U.D.H.; writing—original draft preparation, H.L.W.

**Funding:** This research received no external funding.

**Conflicts of Interest:** The authors declare no conflicts of interest.

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
