# Peer review of "State Financial Losses in Public Procurement Construction Projects in Indonesia"

_buildings, doi:10.3390/buildings9050129_

Round 1

Reviewer 1 Report

The paper presents a research on works which causes financial losses in 64 Indonesian´s construction projects. This topic is relevant for the journal. However, various points need revisions taking into account the following comments:

- Abstract. Last lines. Please add specific results.

- Introduction. There are only 14 references in the whole text. Please expand the introduction section in order promote a deeper analysis of the state of the art. Avoid multi-references (e.i. [2,3,4,5,6]), each reference must be commented.

- Please check if the same criteria is follows for punctuation and if it was done according to the journal template.

- Research method must be expanded in order to explain the statistical analysis

- More specific data must be added in the discussion chapter

- Please define “Rp”

- There are no graphs, in which results are depicted

- Please add the funding if exists, the conflict of interest and authors´ contributions (see the template).

- Check the paper again for any possible misprints (e.i. conslusion)

Author Response

i have improved my journal according to the review

Reviewer 2 Report

1. Table 1 - table header and title - it seems to me that it is a "data set". These are not profiles or criteria. It may be an "area of research" eventually.

2. The calculation of "average value, smallest value, and greatest value" are not the results of descriptive statistical methods. These are the effects of simple mathematical calculations.

Author Response

i have corrected my journal according to the review

Reviewer 3 Report

The title of the paper tells about "Indication of the Types of Work...". More than a half of the chapter 3 (Results) is about location and dates of the failure construction contracts.

Four groups of corruption (lines 32-33) have to be explained (e.g. what is the difference between bribery and fraudulent deeds).

Especially "State financial loses" (mantioned in the subject of the paper too) should be explained. I imagine that it is a difference between planned and real cost, but it is my imagination only. It have to be clear.

The purpose of the paper (line 55) seems to be different than the title give - "to identify the type of construction project..."  vs "Indication of the Types of Work..."

I doubt if "time series" term (line 57) is properly used. I haven't found this type of analysis in the paper.

Lines 60-62

"Research data can be obtained.." They were obtained? or They can be obtained? It is significant difference. I should be clear what was the source of the data analyzed (references not translated).

Again "financial loses" are used which are not defined.

Lines 72-73

What kind of frequency do the Authors mean?

Table 2

What is the reason (sense) of calculating average value of loses per location as number of projects differs from 1 to 13.

There is an assumption made in the paper that state financial loses are clear indicator of a corruption. This kind of a statemet needs a proof (beside defining the state financial loses which is a must).

Recomendation are close to obvious "more attention" "prcedures properly applied" etc.

It is not clear what is included in the references. It seems, there are official acts and reports (it requires translation; even simple e.g. act, ordinance, report. It seems (as there is no translation to english language), there is no reference about financial loses, corruption, way of calculating loses etc.

The hard evidence of state financial loses needs: defining the "state financial loses". Rewriting the paper as a evidence of phonomena would be more appriopriate. Clarifying the difference between kind of works and type of project is necessary too.

Author Response

(The authors gave the same response as above.)

Round 2

Reviewer 1 Report

The authors fulfilled all the requirements. 

Author Response

Response to Reviewer 1 Comments

Point 1: English language and style are fine/minor spell check required

Response 1: the language style have been revised, spelling check has been done

Point 2: Research design could be improved

Response 2: the research design have been updated

Point 3: Conclusion supported by the results

Response 3: the conclusion have been updated

Reviewer 3 Report

Probably instead of "... Infrastructure Projects..."  the phrase "Public Procurement Construction Projects" would relfect the content better. (as table 4 content is: Door, tile work, chair too).

Aanstampingand aKansteen are really not known for me.

My assessment "low merit" arises from...

a/ the paper present the data found by the authors in existing documents (any special analysis is not applied to the data collected). I do not complain the effort of the authors. It could be high even for data collecting.

b/ there are analysis (e.g. published by prof. Flyvbiert) where mega projects are analysed and the conclusion is that roads, briges, rail-roads suffer from cost overruns (even significant) according to poor planning - not neccessaryit is caused by corruption and other illigal processes.

The assumption to the paper was done: cost increase = corruption. As I understand it was made bassd on official reports of Indonesian government and govermental agencies. The paper could be much more valuable if the cost increase is shown as a percantage (ratio) to the value value of a givien project. Then the level of cost overruns could be compared to the level recorded internationally, and then the problem of corruption can be discussed more precisely.

I find the paper as a report of cost increase, but unfortunately the percentage in relation to original values are not presented. So even serwing as a report, the value of the paper is not high.

Author Response

Response to Reviewer 3 Comments

Point 1: Probably instead of "... Infrastructure Projects..."  the phrase "Public Procurement Construction Projects" would reflect the content better. (as table 4 content is: Door, tile work, the chair too).

Response 1: It does make sense for me. The roots of corruption in this case contextually in construction procurement. I agree to revise the phrase. Thanks for your suggestion.

Point 2: Aanstamping and aKansteen are really not known for me.

Response 2: Aanstamping is a term in duch language with a similar definition with ellipse stone as a volcanic stone foundation. The Kansteen have been revised; the right term is curb (the edging of pavement).

Point 3: My assessment "low merit" arises from...

a/ the paper present the data found by the authors in existing documents (any special analysis is not applied to the data collected). I do not complain about the effort of the authors. It could be high even for data collecting.

b/ there is analysis (e.g. published by prof. Flyvbiert) where mega projects are analysed, and the conclusion is that roads, bridges, railroads suffer from cost overruns (even significant) according to poor planning - not necessary is caused by corruption and other illegal processes.

Response 3:

a/ -

b/ I agree about the analysis about megaprojects, but this paper representing symptoms of corruption from the project to raise the issues typology of works, which potentially indicated on corruption.

Point 4: The assumption of the paper was done: cost increase = corruption. As I understand it was made based on official reports of the Indonesian government and governmental agencies. The paper could be much more valuable if the cost increase is shown as a percentage (ratio) to the value of a given project. Then the level of cost overruns could be compared to the level recorded internationally, and then the problem of corruption can be discussed more precisely.

Response 4: The basic assumption is increasing cost. The data of this paper are not officially published. The data is a part of an official investigation on corruption symptom by prosecutors or police.

Point 5: I find the paper as a report of the cost increase, but unfortunately the percentage of original values are not presented. So even sewing as a report, the value of the paper is not high.

Response 5: I agree that the cost value that indicated as corruption is not high. The percentage has been added on the table

Point 6: English language and style are fine/minor spell check required

Response 6: the language has been updated
